# Pressure and temperature induced red-shift of the sodium D-line during HMX deflagration

Olivia J. Morley [1*] & David M. Williamson[1]

The sodium D-line is often present in optical spectra of combustion due to its high prevalence and emissivity. Collision theory predicts the spectral peak to have a red-shift dependent on pressure, $P$, and temperature, $T$. Here we show that the conditions reached during deflagration of octahydro-1,3,5,7-tetranitro-1,3,5,7-tetrazocine (HMX) permit the red-shift of the sodium D-line to be calibrated to 1.5 GPa. Deflagration at these pressures is achieved using a split Hopkinson pressure bar apparatus, with temperatures of *circa* 2900 K from the greybody continuum away from spectral features. Lower deflagration pressures, of 0.5 to 0.9 GPa, are achieved in a fallhammer test, with temperatures of *circa* 4000 K. The red-shift exhibits the predicted $PT^{-0.7}$ dependence with a constant of proportionality of $(950 \pm 30)$ GPa$^{-1}\cdot$K$^{0.7}\cdot$nm. Using the serendipitous presence of sodium, this optical technique allows for fast measurements of both pressure and temperature from the same light source in one measurement.

---

[1] Cavendish Laboratory, University of Cambridge, Cambridge, UK. *email: ojm32@cam.ac.uk

George Liveing, Professor of Chemistry, and Sir James Dewar, Jacksonian Professor of the University of Cambridge, formed a decades long scientific partnership researching the nature of spectral lines culminating in the issuance of their 1915 book of collected papers[1]. Among their many pioneering experiments was a series on the spectral lines of the metals developed by exploding gases[2]. The authors hypothesised that they ought to see distinct blue- and red-shifts of the sodium yellow lines in explosions, depending on whether the explosion was receding from or advancing towards the observer. They were not able to confirm their hypothesis; something they in part ascribed to obfuscation by spectral broadening and the defuse manner in which the lines manifested. They did however remark on how surprisingly bright the sodium lines appeared.

Sodium is a very bright spectrally emissive element whose signature often appears in optical emission spectra of combustion, even for seemingly 'pure' materials where its presence is unexpected[3,4]. At atmospheric pressures, the familiar orange colour is owing to a well-calculated doublet centred at 589.3 nm. Yet in high pressure environments, a small red-shift occurs, which has been noted during sonoluminescencent investigations of salt solutions[5–7]. This particular shift does not have its origin in the Doppler effect, rather it is instead due to the perturbation of electron orbitals by collision. Margenau's early theories on the effect of collisions on atomic spectra, based on differences between the potential energies of an emitter with distance from a perturbing particle, predicted a linear dependence of frequency shift on gas density[8,9]. His complementary experiments into sodium and other alkali metal emissions demonstrated this linear dependence. However, his research was limited to relatively low densities and pressures[10,11]. Later experiments showed this relationship held more generally for alkali metals and a variety of different gaseous perturbers at increased densities[12,13], though in absolute terms the pressures investigated were still relatively modest; of order 10 MPa. A linear dependence is also anticipated by Lindholm-Foley collision theory, shown to be[14,15]:

$$\Delta f \propto \langle v^{3/5} \rangle n \tag{1}$$

where $\Delta f$ is the frequency shift, $n$ the number density of the perturbing gas and $\langle v \rangle$ is the mean molecular speed, derived from the temperature and mass of the perturbing molecule: $\langle v \rangle = \sqrt{(8kT/\pi m)}$. By application of the ideal gas equation, the wavelength shift from collisions is therefore:

$$c\lambda^{-2}\Delta\lambda \approx -\Delta f \propto PT^{-0.7} \tag{2}$$

Thus, the magnitude of the red-shift can be described by a simple function of the measurable temperature and pressure, having a constant of proportionality that is dependent on the interaction parameters of the species involved[16]. Taking the same approach as Fletcher and McDaniel[16], and substituting for the energy levels and polarizabilities (assuming they are constant)[17] of the ground and excited states of sodium, yields a predicted constant of proportionality for the D-line red-shift of $(883 \pm 9)$ $GPa^{-1} K^{0.7}$ nm. This would lead to a measurable shift of order nanometres at GPa pressures and thousands of Kelvin. The standard error derives from the uncertainty in experimentally measured quantities of input parameters, rather the accuracy of the model up to these pressures.

In order to achieve a reasonable function to describe the shift, the calculation takes certain approximations. First, that the perturbing atoms are composed of the reaction products, with properties similar to those of air. This is justified on the basis of $N_2$ being a major product species, along with CO and $H_2O$. Another approximation in the theory is to ignore higher order terms in the dispersion potential, as well as not including quantum exchange interactions. This implies the existence of a high pressure limit to the theory used, which has not previously been explored.

High pressure red-shift can be observed in solid-state fluorescence, where the effect is exploited to calibrate diamond anvil pressure cells; rubies exhibit a linear shift up to tens of gigapascals[18,19]; however, being solid state, this shift does not have its origins in the same collision theory. The red-shift of sodium at gigapascal pressures has not previously been quantified. Sodium emits in its vapour state, and so it is experimentally hard to access the required conditions, especially statically, as the necessary environment asks very difficult questions of the confining materials. Conversely, the intense conditions required can be created with relative ease dynamically; hence, the prior observations of red-shifts from the sonoluminescence community. The difficulty then principally derives from the fleeting timescales involved, and the corresponding demands this places on the diagnostic instruments.

In summary, whereas the theoretical form of the collisional red-shift is known, there is no knowledge whether the assumptions present in the theory will hold under higher pressures, and sodium vapour cannot be easily subjected to these high pressures statically.

In the present study we have elected to create the necessary conditions dynamically by the rapid release of chemical energy during the deflagration of an explosive chemical compound. Deflagration in explosive materials is when the reaction front travels at subsonic speeds (in contrast to supersonic detonation) under moderate confinement, with a degree of self-pressurisation. The material used in our research, octahydro-1,3,5,7-tetranitro-1,3,5,7-tetrazocine (more conveniently referred to as HMX), is a crystalline explosive material that undergoes a violent exothermic decomposition, with an ideal reaction of: $HMX \rightarrow 4CO + 4H_2O + 4N_2$, producing 12 moles of gas from a single molecule, though in reality, the reaction is multi-step and involves a large number of intermediate species[20]. Using optical spectroscopy to determine the pressure from such a reaction has not previously been reported. The HMX molecule does not contain sodium, but sodium is serendipitously present in sufficient quantities in commercial grade HMX as a minor impurity that no adulteration is necessary.

## Results

**Deflagration Spectra.** Deflagration was achieved by two distinct methods. The first was to use a conventional fallhammer apparatus that resulted in ignition at pressures of *circa* 2/3 GPa. The second method utilised a split Hopkinson pressure bar[21] and is, to the authors' best knowledge, the first time such an instrument has been used to successfully study deflagration in HMX, allowing pressures of up to 1.5 GPa to be investigated. In terms of the mechanical loading paths, the main differences between these methods are that the anvils close at a speed of *circa* 4 m s$^{-1}$ in the fallhammer, and between 50 and 75 m s$^{-1}$ in the split Hopkinson Pressure bar. Second, the shape of the force-time histories is an approximate half-sine in the case of the fallhammer, and a top-hat in the case of the Hopkinson bar. The durations of the deflagration events remained unchanged in either case, *circa* 20 μs, and were always significantly shorter than the durations for which the compressive forces were applied, leading to an approximately constant pressure over the time the reaction occurred. The spectrum was measured as an integrated count over the whole reaction, with the optical gate open for 30–50 μs.

In each case the mechanical force-histories showed no obvious deviations from either the half-sine or the top-hat profiles at the time of ignition, as indicated by optical emission, demonstrating that the product gas pressures were in equilibrium with the

applied mechanical forces. There was neither an excess of force owing to a higher gas pressure, nor a fall in force owing to the sudden disappearance of a load-bearing solid component. In the study of Heavens and Field[22], which also made use of a fallhammer-like arrangement, and deployed a photodiode to indicate when ignition occurred, but importantly had no analysis of the spectral content of the light, a corresponding simultaneous drop in force was observed at the point of ignition. However, their anvil configuration had no lateral confinement, and their product gasses were free to vent, precluding self-pressurisation to any degree.

Figure 1 shows typical HMX spectra gathered under conditions of unconfined burning, deflagration during a 0.6 GPa fallhammer experiment, and a 1.0 GPa split Hopkinson pressure bar experiment. These experiments had a level of variation present, such as the width of the sodium line and the relative intensities of the sodium, calcium and greybody emission. Present in the deflagration spectra are both continuous greybody radiation and spectral features. The sodium peak has clearly broadened, with likely contributions from Doppler, collisional and potentially Stark broadening mechanisms. Significantly, the peak is also red-shifted by a few nanometres, with the magnitude of the shift noted to be strongly correlated with pressure.

**Temperature measurements**. The continuous greybody radiation has a shape dependent only on the temperature (Planck's law). Given the 400–500 nm range consistently had no spectral contribution to the emission spectra, a greybody fit over this region

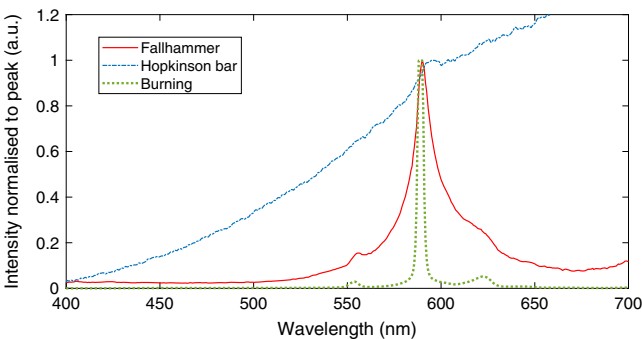

**Fig. 1 Emission spectra for different imposed pressures.** Typical unconfined burning spectrum (green dotted line), 0.6 GPa fallhammer (red solid line) and 1.0 GPa Hopkinson bar (blue dashed line) deflagration. Calcium hydroxide peaks at 555 and 620 nm are also visible in the burning spectrum, alongside the main 589 nm sodium peak.

can determine the temperature of the reaction (Fig. 2). For the fallhammer, such a fit resulted in temperatures of 4100 ± 300 K across the experiments. These values are in agreement with previous measurements[23], and the magnitudes expected on the basis of a thermodynamic consideration of the exothermic reaction[20].

Somewhat unexpectedly, the Hopkinson bar arrangement gave lower temperatures of 2900 ± 200 K across all measured pressures. The spectra are also more blackbody-like, with less-prominent spectral features. We postulate that the different mechanical loading paths, and the higher imposed pressures, may be driving different reaction pathways, and the formation of final products in differing proportions. For example, a higher external pressure may favour solid reaction products, such as soot[24], which would then provide a higher number of blackbody emitters relative to a fixed number of sodium emitters, diminishing the spectral appearance. At the same time, the incomplete oxidation of carbon may decrease the heat of the reaction, and account for the lower measured temperatures.

**Red-shift**. From each spectrum, the centre of the sodium peak was measured, and the red-shift calculated. This was then plotted against pressure and temperature in the functional form given by Eq. (2), (Fig. 3).

## Discussion
As can be seen, the data are consistent with collisional impact theory,

$$\Delta\lambda \approx 950\,PT^{-0.7} \qquad (3)$$

with a constant of proportionality equal to (950 ± 30) GPa$^{-1}$ K$^{0.7}$ nm using linear regression and explicitly taking account of the uncertainties associated with each individual data point. Considering the approximations made in calculating the theoretical value of *circa* 880 GPa$^{-1}$ K$^{0.7}$ nm, this is in good agreement with the Lindholm-Foley description, and follows the same functional form. At the higher pressures, a potential upward curve (deviation from linear) can be seen, though it is still within error, which may hint at the start of the high pressure limit to the theory. Regardless, for the range of pressures presented, Lindholm-Foley collision theory is still a good model for the behaviour of red-shift with increasing pressure.

It is worth considering what other mechanisms could be present, such as the Doppler and Stark shift, that could cause a red-shift in spectral peaks. However, by examining the system one can see that these will be relatively insignificant effects. For the Doppler effect, the emitting particles would have to all be moving away from the receiving fibre, which is unphysical. In addition,

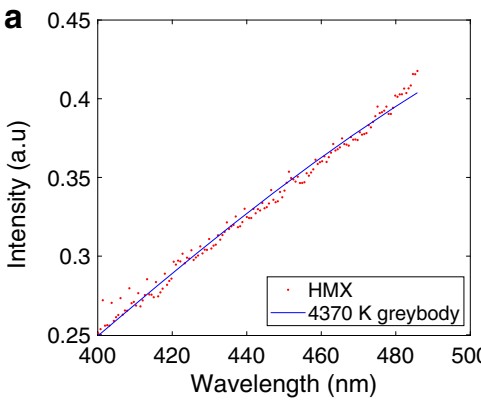

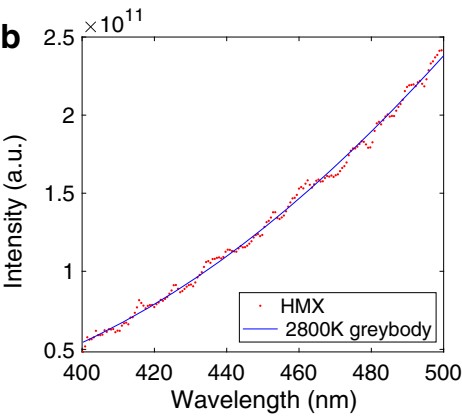

**Fig. 2 Greybody temperature fits to spectra.** Typical greybody fits to determine temperature applied to non-spectral portions of the fallhammer **a** and Hopkinson bar **b** spectra.

the largest shift one might expect (if the sodium atoms are moving at the detonation velocity of $8\ km\ s^{-1}$) would be of order 0.02 nm. In terms of the Stark shift, based on detonation measurements[25], an electric field of order $10\ kV\ cm^{-1}$ can be estimated to be present in the reaction. This would, given previous measurements of the DC Stark shift of the sodium D-lines[26], result in a shift of order $2 \times 10^{-5}$ nm. Even taking account of ionisation of reaction products, the Saha-Boltzmann equation would predict an electron density of *circa* $5 \times 10^{15}\ cm^{-3}$, with an associated Stark red-shift[27] of order $10^{-4}$ nm, which is negligible compared with the collisional effects. It follows therefore, that Lindholm-Foley theory alone is sufficient to describe our experimental data, correctly describing both functional form and magnitude, and that upon consideration, alternative mechanisms that are known to produce colour-shifts, are negligible in comparison.

The calibration being now known, it affords a simple and novel method to calculate the instantaneous pressure and temperature of the reaction, provided the existence of both greybody and spectral features, in a manner analogous to a ruby-calibrated diamond anvil cell. Previous reviews on lower pressure collisional shift[28] were inclined towards more astrophysical applications, and this research shows that such a mechanism can be expanded to more diverse fields. Whether in sonoluminescence, the science of energetic materials, or perhaps exploited in an optical pressure sensor for industrial applications, this will enable pressure measurement at the exact point and time of emission as experienced by the sodium atoms.

Returning to the experiments of Liveing and Dewar, they estimated that the conditions of their explosions were of order 3000 K and 10 atmospheres of pressure. Could they have observed a collisional red-shift in their spectra? Maybe! Based upon our new results, the corresponding sodium red-shift for their conditions would be of order 3 picometres. From the description of their instrument, their resolvance was also of order 3 picometres. In the case of another alkali metal, lithium, about which they provided the greatest detail, in their own intriguing words; 'certainly there appeared to be a very slight displacement, but it was not so definite that one could be sure of it'.

## Methods

**Optical equipment.** During experiments, the light from a reaction was directed via optical fibre into a 50:50 beam-splitter [ThorLabs CM1-BS013], which sent half the light to a single-shot gated spectrometer [EG&G Princeton Applied Research Models modules 1235, 1304, 1455, 1471], and half the light to a photodiode [EOT ET-2030] connected to an oscilloscope [Tektronix DPO 2024], allowing both the wavelength spectrum and time intensity profile of the light to be measured. A spectral transmission calibration for the system was conducted using a lamp with a known blackbody temperature (NPL certified), and the ratio between this measured spectrum and the expected blackbody emission was used as a prefactor $K(\lambda)$ to experimental spectra, to account unequal transmission in the usual manner. The spectrometer has 0.6 nm resolution over the entire visible range, and 0.06 nm resolution over a window covering only the spectral features owing to a choice of gratings with different line densities (150 or $1200\ g\ mm^{-1}$).

The optical gate was open for 30–50 µs, in order to catch the majority of light from the reaction (Fig. 4).

**Material.** $\beta$-polymorph HMX with the Type-B[29] particle size distribution (ROF Bridgwater, $\beta$-polymorph octahydro-1,3,5,7-tetranitro-1,3,5,7-tetrazocine). The material was oven-dried at 110 °C for 24 h prior to use. All experiments were conducted with material from the same batch.

**Open burning.** The spectral details of HMX burning unconfined at atmospheric pressure were obtained by simply burning the material in an open butane flame. This produced an optical spectrum that was predominantly spectral, with peaks from the highly emissive alkali metal impurities and their compounds dominating in all measurements taken. Especially prominent was the sodium doublet, making its customary appearance at 589.3 nm.

**Deflagration in the fallhammer apparatus [OZM Research. BFH-10 BAM].** A loose-packed volume of $20\ mm^3$ of HMX was placed between a pair of 10 mm

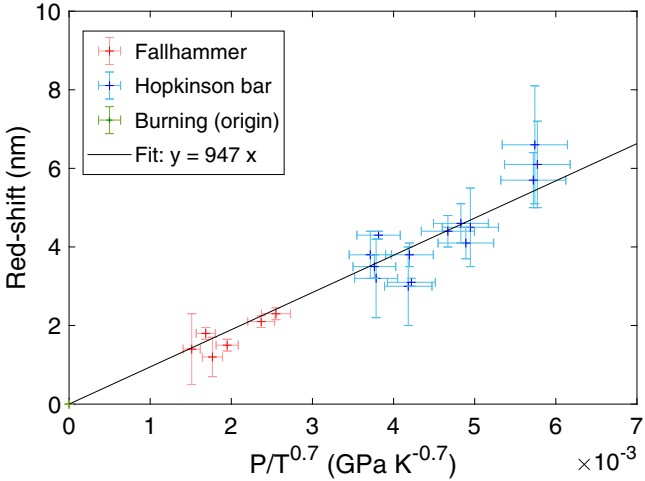

**Fig. 3 Red-shift measurement.** The relationship between the measured pressure and temperature, and red-shift, of the sodium doublet across the three regimes of burning (green at origin), fallhammer (red) and Hopkinson bar (blue) deflagration. Vertical error bars show the uncertainty of peak centre, and horizontal errors were calculated from the uncertainties in pressure and temperature measurements, with the assumption these were random uncoupled errors, and therefore could be combined in quadrature.

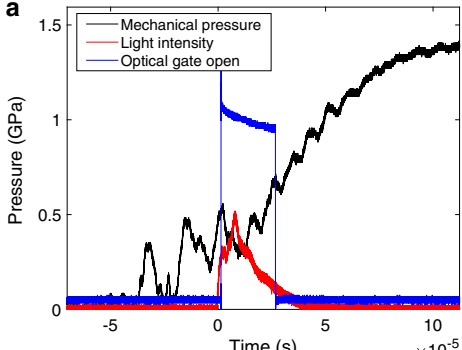
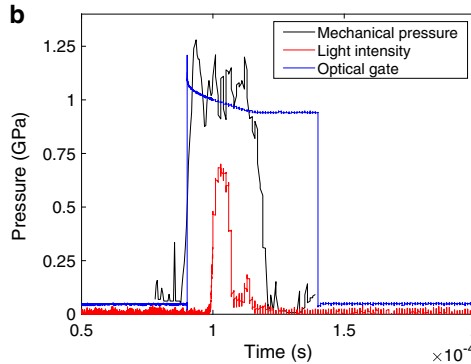

**Fig. 4 Pressure-time histories.** The measurement of the mechanical pressure applied against time, with the time of reaction (light from photodiode) and when the optical gate of the spectrometer was open (when the signal is high) for the fallhammer **a** and the Hopkinson bars **b**. Hopkinson bar pressure data time shifted to correct for delay caused by propagation of the pressure pulse from the specimen to the gauge station.

diameter hardened steel anvils, confined by a cylindrical annulus of hardened steel, onto which a 2 kg weight fell from a 90 cm height, chosen to cause a deflagration reaction each time. This provided a mechanical force-time profile reminiscent of a half-sine with amplitude *circa* 1.4 GPa and duration *circa* 300 μs. Note that deflagration would typically occur well before the peak mechanical pressure was reached.

The enclosing cylindrical annulus had a wall thickness of 20 mm and two diametrically opposing radial ports having their centres coincide with the plane of the explosive sample to provide access to the deflagration. Into one port, an optical fibre was tightly fitted to carry the emitted light to the beam-splitter. Into the second port, a dynamic gas pressure gauge was fitted [Kistler 6215]. The upper of the two steel anvils was instrumented with a semiconductor strain gauge [Kulite Semiconductor Products, Inc. AFP-500-090]. Both the Kistler and the strain gauge were calibrated so as to allow for time-resolved measurements of reaction product gas pressures and the mechanical forces acting on the system. These measurements were strongly correlated and in almost complete agreement, after accounting for gas expansion into the dead-volume of the Kistler gauge; corroboratively indicating that the reaction product gases are in pressure equilibrium with the mechanical forces supplied by the changing momentum of the falling weight as it is arrested by the impact.

**Deflagration in the split Hopkinson pressure bar**. A split Hopkinson pressure bar is in essence a series of waveguides for elastic waves[5]. All bars were 12.7 mm diameter and fabricated from maraging 350 steel. A striker bar of 150 mm length was incident upon an input bar of 450 mm length at speeds of between 50 and 75 m·s⁻¹, generating top-hat shaped compressive pressure pulses of duration *circa* 60 μs and corresponding amplitudes of *circa* 1.0 and 1.5 GPa respectively. These propagated along the input bar and were incident upon the specimens that were located between it and the 450 mm length output bar. Some of the energy of the wave was reflected from the interface and some is transmitted through and propagated along the output bar, and into a momentum trap formed of a 200 mm length of bar that was free to move and be gently arrested. In operation the system acts somewhat like a Newton's cradle.

The particle velocities associated with the passage of the elastic waves in the input and output bars were measured using a Photon Doppler Velocimeter[30]. Particle velocities are readily converted to mechanical pressure if the mechanical impedance of the bar materials are known; for our maraging 350 steel bars the experimentally measured value is 39,483,160 kg·m⁻²·s⁻¹.

A loose-packed volume of 40 mm³ of HMX was used for these deflagration experiments, again confined by a cylindrical annulus of hardened steel, wall thickness 10 mm, with a single radial port having its centre coincide with the plane of the explosive sample to provide access to the deflagration. Into this port an optical fibre was tightly fitted to carry the emitted light to the beam-splitter. We found that the explosive powder could be spread evenly over the input/output interface by having the confining sleeve in position, and whilst applying light pressure, holding the input bar stationary and giving the output bar one full rotation. Deflagration was observed for applied mechanical pressures greater than 1 GPa, and spectra were recorded for input mechanical pressures of between *circa* 1 and 1.5 GPa.

**Calculating the theoretical shift**. This method was taken from Fletcher and McDaniel's paper[16], using the original theory developed by Traving[14].

The shift expected in the impact regime of Lindholm-Foley collision theory is given by:

$$\Delta f_s = \frac{2.94}{2\pi}(\Delta C)^{2/5} v^{3/5} n\left(\frac{\Delta C}{|\Delta C|}\right) \quad (4)$$

Where $\Delta f_s$ is the frequency shift in Hz, $n$ includes all perturbers and $\Delta C$ is defined as:

$$\Delta C = \frac{3E_p}{2h}\frac{\alpha_p^0}{(4\pi\varepsilon_0)^2}\left[\frac{E_{A,L}\alpha_{A,L}^0}{(E_{A,L}-E_p)} - \frac{E_{A,U}\alpha_{A,U}^0}{(E_{A,U}-E_p)}\right] \quad (5)$$

Where $E$ refers to the ionisation energy of the perturbers ($E_p$ – using the value in air of 14.86 eV), and the sodium atom ($E_{A,L}$ for the lower energy state and $E_{A,U}$ for the upper). Similarly, the $\alpha^0$ refers to the polarizabilities of the states. So for this research into sodium, the energy levels and polarizabilities for the D2 sodium lines[17] were used, instead of the iodine values presented in their paper.

Using $v^{3/5} = 0.891\, v^{3/5}$, as well as the ideal gas equation and speed of the perturbing molecules, leads to a shift of:

$$\Delta f_s = -2.18\times10^{23}(\Delta C)^{2/5}\frac{P}{T^{0.7}}\ \text{Hz} \quad (6)$$

and a wavelength shift, for a sodium at 589 nm, of approximately:

$$\Delta\lambda_s = 2.51\times10^{20}(\Delta C)^{2/5}\frac{P}{T^{0.7}}\ \text{GPa}^{-1}\ \text{K}^{0.7}\ \text{nm} \quad (7)$$

Using the calculation of $\Delta C$, this then produces $\Delta\lambda_s = 883\frac{P}{T^{0.7}}\ \text{GPa}^{-1}\ \text{K}^{0.7}\ \text{nm}$.

**Error analysis**. For the theoretical value, the uncertainties in the values used in the calculation were brought forward. Approximations from the theory were qualitatively stated, but not quantified, as the purpose of this paper was not to provide a rigorous quantitative description of the underlying theory.

In the calculated gradient of Fig. 3, linear regression on the points alone led to an uncertainty of 70 GPa⁻¹ K⁰·⁷ nm (and 50 if weighted using the error bars provided). Constraining the fit to additionally pass through the origin, from the physical expectation, resulted in the measured error of 28 GPa⁻¹ K⁰·⁷ nm, presented above as 30.

## Data availability

The spectra, Photon Doppler Velocimeter, photodiode intensity and pressure gauge data that support these findings have been deposited in the Apollo repository (https://doi.org/10.17863/CAM.41675).

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

## Acknowledgements

O.M. was supported by Emmanuel College, Cambridge, and the Avik Chakravarty Memorial Fund for Physics. D.W. would like to thank AWE for financial support. This research was partially supported by Defence Ordnance Safety Group, Science and Technology.

## Author contributions

O.M. and D.W. equally contributed to this research.

## Competing interests

The authors declare no competing interests.
