## [Peer Review File · Communications Chemistry]

Reviewers' comments:

Reviewer #1 (Remarks to the Author):

Does the manuscript present original and timely results that significantly advance the knowledge in the field?

Yes.

Does the manuscript report on convincing and rigorous data, methods, and analysis?

Yes.

Is the manuscript clearly written in correct English, well organized and free from ambiguities?

Yes.

Is the title descriptive of the contents, concise, interesting, and free of acronyms?

Yes.

Does the abstract adequately and clearly describe the contents (problem, approach, findings) of the paper?

Yes.

Are the figures in the manuscript necessary, adequate, well presented and clearly labeled?

Yes.

Is the reference list appropriate?

Yes.

The authors provide an interesting look into dynamic pressure/temperature events using what is essentially a spectator atom in the form of impurities which naturally exist. The contrasting techniques show an intriguing difference in behavior which is well captured by the emission spectra. Interpretation of the spectra is well supported in historic literature and the fits to the data appear reasonable.

The authors should make a statement of reproducibility in these experiments. Deflagration events tend to be fairly stochastic, especially with fallhammer tests. It would be interesting to note if different pressures/temperatures were observed based on subtle changes such as the packing of the HMX which are not strictly accounted for.

As the authors describe, pressures in both the fallhammer and Hopkinson bar used are transient, time-dependent things. The authors describe mechanical force-time histories as a half-sine and top-hat respectively, but do not fully describe the time-domain over which the spectra and thus temperature are recorded. The authors should state for both experiment types how long the nominal peak pressures last as well as how constant they are, given that the measurements are pressure dependent. This should include where along the pressure trace the spectra/temperature are measured and how long the optical gate lasts. Together this provides an understanding of the state of the sample during measurement in an otherwise dynamic experimental arrangement.

m.s-1 should be corrected to m·s-1 in many places.

The authors should qualify the statement that the Hopkinson bar-produced spectra are "more blackbody-like". Is this a statement of less statistical noise in the fitting of the model or a statement of the existence of fewer well-described deviations from the model such as emission peaks.

It is unclear how the graybody spectrum is deconvolved from the sodium emission, which leads to questions about the color temperatures of the spectrum. In the case of the fallhammer, the emission in figure 1 looks to be simple baseline noise over the region fitted for temperature. The authors should show how the spectra are deconvolved and fit the deconvolved spectrum. Alternatively, rather than fitting to different spectral ranges for the two experiments, the authors could simply write off the 550 – 650 nm band as heavily obscured and fit both the 400 – 550 and

650 – 700 nm spectral band. Adding the red region to the fit region would greatly increase confidence, especially in the lower temperature fits.

Could the broadening of the sodium line be, at least in part, a function of the temperature and pressure spread rather than the absolute peak? Given the confinement and pressure fields in the experiment, it is likely that a substantial temperature gradient exists in the sample which would cause a broadening in the sodium emission, as different volumes are exposed to differing conditions.

While the data is certainly compelling in the GPa pressure range, to extrapolate down to the 10⁻³ nm (pm) red shift regime for correlation to the experiments of Liveing and Dewar seems to be stretching the validity of the model.

Reviewer #2 (Remarks to the Author):

Please see the file attached.

Reviewer #3 (Remarks to the Author):

The work is interesting and relevant to those working in the field of high explosives and provides a novel new diagnostic for determining pressure and temperature through measurement of the red-shift. As the authors note it is somewhat surprising that the SHPB gave lower temperatures across all measured pressures. While the authors offer some supposition for this, it would be worth further investigation to support the observation.

While the authors make a linear fit to the red-shift versus $P/T^{0.7}$, the data appears to be turning up at the higher end of P/T measured, albeit within the rather large error bars. If the goal is to provide a calibration between pressure and temperature, and red-shift, it would be worth extending to higher values of P/T to show whether it is truly a linear relationship or where it breaks down. This could potentially involve other classic DDT test platforms and would make the work much stronger.

The authors make a point of this being the first work employing a SHPB to investigate deflagration of HMX. Extensive work has been performed with the SHPB on explosives, including HMX based explosives, to study the high strain-rate mechanical stress-strain response of these materials. In general, the goal for SHPB work on explosives has been explicitly to avoid the regime on initiation and work has focused on formulations where it is difficult to obtain sufficient sustained pressure in the SHPB to obtain reaction. While there are examples of explosives reacting under SHPB loading they have typically been seen as failures, rather than the focus of the experiments. An intermediate study towards reaction in a SHPB can be found in V.S. Joshi "Recent developments in shear ignition of explosives using hybrid drop weight-Hopkinson bar apparatus" AIP Conference Proceedings 955, 945 (2007); <https://doi.org/10.1063/1.2833285>. The novelty here is that the authors are studying a loose compact of HMX without binder and is therefore sensitive enough to react in the SHPB loading regime.

In Figure 3 the datum for the burning case at the origin is very difficult to make out and may be worth labeling or otherwise make it more obvious to the reader.

Review of "Pressure and temperature induced red-shift of the sodium D-line during HMX deflagration" by Morley and Williamson.

The manuscript describes a study of collision-induced line shift of the resonance sodium line (589 nm) by neutral particles in the environment of HMX deflagration. In a number of experiments using fallhammer and Hopkinson bar setups, the line shift was measured, alongside with the temperature inferred based on the greybody spectrum fits in a spectral region supposedly free from discrete lines.

It is stated that the experimental results excellently agree with the theoretical prediction of the P- and T-dependence of the shift.

The authors believe this is a first study of this kind, and expect the outcomes to be important for other research, from astronomy to industrial applications.

I would like to understand why the authors see importance of this study. Let me explain. First, they calculate a value theoretically, then measure it. Both values are claimed to be accurate to within about 1%, and agree nicely. This is an outstanding result, but somehow the authors do not appear surprised. It means they really trust the theoretical calculations to be 1% accurate. If so, why one needs an experimental verification? The Lindholm-Foley theory is almost a century old, and there have been many studies employing it in one way or another, but to the best of my knowledge, such an excellent agreement has been neither reported nor - more importantly - expected. I add that most studies focus on the line width, not shift; the latter is more challenging both for calculations (e.g., cancellation effects) and measurements (some issues are outlined below).

Regarding the claimed importance of the results in application to other fields, in particular astronomy. It needs a justification. Namely, existence of astrophysical objects/regions for which the neutral broadening is dominant, as is supposedly the case in the present study. Otherwise, the much stronger Stark broadening due to the charged particles, electrons and ions, would dominate the line width, rendering the present results largely unimportant in that context. Furthermore, often, the collision line shift cannot be distinguished from the Doppler shift in a straightforward way. It is not an easy task for a

laboratory
study, and even more so for astrophysics.

To make sure there is no misinterpretation of the result, one should provide a cross-check. For example, to investigate the line `_width_` (based on the same Lindholm-Foley theory; by the way, the Lindholm's name is misspelled in the manuscript). This is not a bullet proof, since the Doppler effect may result in the line broadening, too, but at least this would be a minimal verification. Unfortunately, the authors have not done it, and even do not discuss. Why?

Rather surprisingly, there are no details on the calculations. The claimed accuracy of the theoretical shift value - 1% is even more surprising. The approximations assumed in the Lindholm-Foley is one thing. The other is the uncertainties in the chemical composition of the "bath". Is it really known within 1%? The line shift depends on the thermal velocity, and this velocity (even if the temperature is known exactly) depends on the molecular/atomic mass of the particles causing the broadening. More precisely, it is the `_relative_` velocity of the perturber-radiator system. Again, what was assumed as a set of (neutral) components present in the deflagration of HMX?

Furthermore, there is no discussion of the Stark broadening. The temperature, especially in the case of the fallhammer data, is relatively high, so probably there is a certain amount of ions and electrons present. The charged particles typically cause a two-order-of-magnitude stronger broadening than neutrals, so for the claimed 1%, the ionization degree should be below $1e-4$. Can it be proven?

The accuracy of the experimentally determined shift constant (~1.5%), is similarly questionable, in my opinion. Looking at Fig. 3, I would say the uncertainty is about 10%. Please provide a proper statistical analysis that proves the claimed 1.5%.

Let us now come to the interpretation of the measured spectra. In addition to the resonance Na line, there are (Fig. 1) two more weaker lines attributed to calcium hydroxide. Could there be other Ca lines, perhaps right or very close to the position of the Na line? Looking at the NIST data

I can see a bunch of other Ca lines of comparable intensities in the region of interest. In fact, some of them originate from from excited levels that lie lower than the upper level of the 551-nm line, and thus, should be stronger. Say, the Ca 585.745-nm line,

which would be unresolvable from the Na line, I believe. Needless to say, if present, it would alter the interpretation of the line shift.

Also, what are the small periodic structures seen in the "Fallhammer" data on Fig. 1?

Dear Reviewer 1,

Thank you for your time and consideration in reviewing this manuscript.

Before we go point-by-point replying to your queries, we would first like to draw your attention to the fact that, after further analysis in the interim (and taking account of an offset present we had not previously accounted for), we have re-calibrated the temperature involved to result in slight difference to those previously reported. For that reason, the Hopkinson bar temperatures are now reported as 2900 K, rather than 2500 K, and the gradient has changed to 950 as a result. This has given us the opportunity to re-do all our calculations and we are satisfied everything is now correct.

1. Comment on stochastic nature and reproducibility of experiments, and any changes with the packing of HMX.

It is well known that these experiments have a lot of natural variation that cannot be controlled [Tommy's paper], however every experiment in my research that was conducted with all the diagnostics mentioned have been included in the results.

In general, there was a large variation in the spectral line widths, and relative intensities of the species involved. However, the 'shape' stayed similar, and the position of the peak reasonably constant for each regime. The temperature was also fairly similar between different measurements, as reflected in the standard deviation given with the mean result. We have added comments along these lines in the manuscript.

The largest variation was the line width, which did not strongly correlate to anything (understandable, given the number of factors that might contribute to broadening, including the fact this is a single shot integrated spectrum), and whether Ca was visible in the spectrum or not.

We did not try different packing (we stuck only to a constant volume), but this is certainly an area that would be of interest in further research. As would a number of different variables – such as quantity of sodium, energetic material (e.g. RDX) used or particle size distribution. The aim of this paper was to highlight the behavior of sodium naturally present in energetic materials and its uses in a general sense, rather than focusing on other variables - that would be another paper's worth of results!

2. Time of top-hat and half-sine. How long peak pressure lasts and how constant they are.

We have added to the manuscript typical traces of the pressure, along with when the reaction occurred in the Methods section (see figure 4). For both, the reaction was fast enough that the pressure was near constant.

The optical gate was open over the majority of the reaction itself, lasting for tens of microseconds. This was to ensure the whole reaction was included. As long as the entire reaction (excluding a small amount at the start which triggered the gate to open) was covered, keeping the gate open another few tens of microseconds did not, to my knowledge, affect the results (as no more light was being emitted) due to it being an integrated experiment, however it was kept as short as reasonably possible to avoid stressing the MCP as much as possible.

3. What is meant by 'more blackbody-like'?

Less sodium emitters relative to blackbody – less spectral peak deviation. Instead of seeing an Na peak with some background emission, we saw a continuous increasing emission with a small peak in it. One would expect the number of sodium emitters to be similar between experiments (as the material used was the same), so we theorize there are more blackbody emitters present. We have specified this in the revised manuscript.

One would expect the number of sodium emitters is fixed per unit mass of material; number of blackbody emitters (chiefly solid carbon) is dependent on how the reaction proceeds and vary with experimental conditions. If Na/C is high, expect spectral emission, if Na/C is low, expect a more blackbody emission.

4. Deconvolving the spectra and fitting to different bands.

We found trying to deconvolve the spectra difficult. Each measurement had a different line width, and the relative presence of Ca changed each time as well. Added to the fact that it is an integrated measurement meant fully deconvolving sodium became impossible.

Notwithstanding the above, we found that the 400 – 500 nm band was free of spectral emission in the fallhammer at all times, and used this band to gain temperature measurement. For a fair comparison, this was also the band used to determine the temperature in the Hopkinson bar. 550 - 700 nm were affected by the sodium, and calcium, emissions and data from this region was therefore excluded from the fits.

We agree that a fit to the other side of the peak would increase confidence. However, this held two major problems. Firstly, the sensitivity of the spectrometer is much reduced after 650 nm, and by 700 nm the number of counts in the raw data is low. Secondly, is the fact that potassium emission (centered around 770 nm, but presumably also broadened) does also affect this region. Potassium is normally not seen as it is far into the insensitive region of the spectrometer. The fact we have recorded it in some burning emission spectra indicate it is likely to be as emissive – if not more so – than the sodium.

Therefore, the temperature measurement method is limited to the range between 400 and 500 nm.

You are correct in mentioning figure 1 shows a spectrum with a high sodium emission relative to blackbody emission. This figure has been normalized and is intended to emphasize the fact that the peak has shifted. The raw data, before calibration, for the 400 - 500 nm region contains counts significantly above the noise level.

When extracting the temperature, the method focuses onto the region of interest (400 -500 nm), and examines the data. The periodic oscillations seen in the previous figure 1 are larger artefacts from the spectrometer and the calibration method (after taking the majority of the measurements, the spectrometer developed a habit of dropping the intensity periodically on every fourth pixel). The new manuscript has adjusted figure 1, removing these periodic artefacts, as was done before making any temperature measurement quoted here.

[REDACTED]

5. Broadening of sodium due to temperature gradients

Broadening is a lot more complicated than shift, given all the mechanisms present that can cause it. Fortunately, it is a symmetric process, so should not affect the red-shift.

Interesting, the time-resolved measurements show a roughly constant temperature throughout the reaction – see the figures above.

6. It's a stretch to apply it down to 3 pm

We absolutely agree. It's intended more as a fun observation – a comment on the historical paper – rather than trying to prove the theory. We are not wholly convinced the authors saw something definitive – as they themselves admitted.

I hope this answers the questions you had. Please let me know if you require anything else, or have a suggestion to make – as an expert in your area, your queries/suggestions are always appreciated.

Dear Reviewer 2,

Thank you for your time and consideration in reviewing this manuscript.

Before we go point-by-point replying to your queries, we would first like to draw your attention to the fact that, after further analysis in the interim (and taking account of an offset present we had not previously accounted for), we have re-calibrated the temperature involved to result in slight difference to those previously reported. For that reason, the Hopkinson bar temperatures are now reported as 2900 K, rather than 2500 K, and the gradient has changed to 950 as a result. This has given us the opportunity to re-do all our calculations and we are satisfied everything is now correct.

1. Importance of this research – why do the experiments?

This is, first and foremost, experimentally driven research. In the first instance, we had not anticipated the sodium line to appear at all, thereafter noted it was in the ‘wrong’ place and then wanted to characterize and explain it.

We read about the possible reasons for such a red-shift, and concluded that the Lindholm-Foley description was most plausible/feasible, given the conditions present. We then tested this by applying higher pressures, and using another piece of equipment, and saw it followed the correct functional form – indicating that L-F theory still provides a good description of the physics under these circumstances.

As you correctly point out, the theory has many approximations; for example, we had to go with ‘idealised’ reaction, and made the assumption that the perturber atoms would behave similar to air, though a primary reaction product is N_2 and H_2O , so not a large deviation. Also, to the best of our knowledge, the shift has never before been measured in such an experiment. Given the system is so complicated, there was no reason to believe, *a priori*, that L-F theory would hold under these circumstances (pressures). Now we know.

This makes it important to conduct these experiments – to see if such an approximate theory still can be used in these extreme environments, far from those originally envisioned by Lindholm and Foley.

We used the Lindholm-Foley description as it had a combination of:

- Encompassing the physics and being comprehensive enough to capture the essence of what is occurring
- Accessibility and ease of calculation, including having input parameters such as energy levels and polarizability.

The inclusion of the calculated value is an indicative value of the expected shift if the theory still holds, calculated following Fletcher’s paper (more on this point below). The error stated is the uncertainty in the output of the calculation on the basis of the uncertainties on the input parameters. It does not relate to anything intrinsic to L-F theory itself, we are not seeking to refine or extend L-F theory, only discover the limits of its applicability. An emphasis we did not make clear and have now adjusted in the manuscript.

You mentioned this is not a well-researched area – we very much agree with you here! Previous research into this theory does tend to focus more on the line width. We did not take line-width into consideration as we wanted to:

- Principally, investigate the shift I'd seen in the experimental data and see whether there was a theory that could describe it
- Focus on a not-well-researched area. This shift is not well known outside of astrophysical applications.

Overall we would say part of the importance of this research – other than the benefits of optical based pressure measurements – is to bring attention to this shift. We cannot speak for every spectroscopist, but there is a lot of interest in this method from those in the energetic materials community – one example is given by Reviewer 3; using this diagnostic to look at DDT (deflagration-to-detonation transition) would show how the pressure changes along this reaction.

It is a chance to bring a piece of theory into the knowledge of experimentalists, so we can use it to perform measurements previously not conceived of.

2. Importance in astrophysics

We did not provide specific examples for the uses in different fields - the aim of the paper was to keep it general and as inter-disciplinary as possible. We mention astrophysics specifically as the majority of papers we read on the shift were written by astrophysicists as it is pertinent to their observations. The review written by Ch'en and Takeo was inspired, we believe, to review the effect of pressure in regards to astrophysical applications.

The shift has historically been more of a 'known' effect in astrophysics rather than in other communities, such as energetic materials. Therefore, for historic reasons, we do think the manuscript is important to that community, as it continues and adds to the corpus of research dating back decades, in particular by extending the regime of applicability.

3. Conditions where neutral line broadening is dominant

The research presented here is very much focused on explaining the presence of a line shift, rather than investigate the broadening present.

Like broadening, there are other shift mechanisms. The Doppler shift and Stark shift are also present, though these are less dominant in this experiment. We added a paragraph into the paper with some rough order-of-magnitude estimations to show why we are not concerned in my case with these effects.

Each scenario is different, and so there may be circumstances in other classes of experiments where these shift mechanisms are non-negligible compared to the collisional one. Yet, unlike broadening, the collisional mechanism for shift is dominant in the regime investigated.

4. Can't be distinguished from Doppler shift

In systems where the Doppler shift is non-negligible – such as astrophysical ones – it will then have to be considered more carefully.

The Doppler shift is a result of the velocity of the emitter, and so equal for all emitting species. In contrast, the collisional mechanism depends on the energy levels and polarizabilities of each emitter, and so will result in different shifts for different elements. In a spectrum containing multiple spectral lines, which is often the case for astrophysical measurements, it can therefore (theoretically) be separated.

5. Investigate line-width

The focus of this paper was on the shift, independent to the line width. Measurements of the line width are rather complicated in this setup. Given the environments present we would expect temperature, pressure and potentially (as you said earlier) Stark broadening to be involved. Adding to this, the spectrometer integrates over a period of time, and there are other spectral peaks inside the sodium one, means using the line width becomes very complicated.

We did extract line-width from my experiments, however we could not correlate these values with any physical meaning, and found it difficult to de-convolve them. Instead, wrote the paper based on shift alone, for which the explanation is far more straight forward with no loss of utility.

Investigating the line-width would not, in these circumstances, provide details on the reaction not already measured. It would be a more stringent test on the model, but that is not the motivation here.

6. No detail on calculations

We could provide more detail; however, as we said above, it is using the method entirely given in Fletcher's paper, and we have limited space to show it in detail and chose not to avoid needless repetition. We have now included some stages of this calculation in the 'Methods' section.

As this was intended to confirm whether Lindholm-Foley collision theory was the likely explanation for the experimentally measured values, we did not add anything original to the maths, we simply re-performed the calculations inserting the experimental values for sodium in the place of those of iodine.

The uncertainty is a result of the uncertainties of the constants used in such an equation – not the uncertainty given the approximations assumed by the theory. We would like to thank you for your comments – the uncertainty was not presented in such a way. As you point out – the composition of the 'bath' of chemicals is not as well known, also that this model has multiple approximations in built, and so cannot provide a certain level of accuracy if you wanted to model independent of experiments.

To that end, we have mentioned what the uncertainty refers to specifically, with a longer description of the approximations present.

7. No discussion of Stark broadening.

Related to above, there is little discussion of any broadening, as that was not the intended purpose of the manuscript. We have included it as a possible mechanism for the broadening, as well as mentioning the Stark shift as another shift mechanism. This relies on an estimate of the electric fields present in the reaction – for which we used an experiment on detonation (reported in the literature and cited) to get an order-of-magnitude estimate, which is negligible compared to the collisional effects.

8. Check errors on gradient

We did straight line regression with the usual error analysis. When taking *only* the data points into account, we ended up with an error on the gradient of around 8 % - as you would suggest. There are other considerations that one can then apply.

For example, constraining the line to go through the origin, which will lessen the uncertainty by removing the variable intercept value. Also a weighted mean – taking the magnitude of the uncertainty at each point into account – was also applied.

We included a short explanation of this in the Methods section, with the errors given using each approach. We have not gone into detail on the equations used, as they take up a lot of space and are not novel.

<https://www.che.udel.edu/wp-content/uploads/2019/03/FittingData.pdf> (A. D Shine, 2006) is a pdf that goes into detail on the different uncertainties (it is written for engineers, but *does* go through the different types of error analysis that can be used).

I've attached my data and analysis in the form of an image of an excel spreadsheet with the points and calculations on (at the end of this response as the submitting system did not allow excel format).

9. Other possible spectral lines

The only spectral lines seen belonged to sodium, CaOH, CaO and potassium (located at 770 nm). No other lines were every detected.

The spectrometer has 2 settings, a wide-range (as shown in figure 1) with a resolution of 0.6 nm. We can also do a higher resolution scan (0.06 nm) over a smaller range.

We did the higher resolution scan over the sodium doublet to confirm it was sodium at the beginning of this research. We can provide the spectrum, with the resolved double peak, if required.

The only lines we ever saw (in both burning and deflagration) were the sodium at 589 nm, potassium at 770 nm and the collection of CaO and CaOH lines at 555 and 620 nm. We took multiple spectra of burning HMX, and whilst the intensities of Na and Ca did vary (sometimes no Ca was seen as the Na completely dominated), no other lines ever appeared.

This is in agreement with other combustion measurements. Sodium appears in practically every flame spectrum, as Maxwell, K. L. & Hudson, M. K. (citation 4 in the manuscript) said, 'One can note the sodium line emission at 589 nm as a sort of "landmark" for this study, as it is always present '.

The CaO and CaOH lines at 555 and 620 nm are similar, though not as emissive. In Gaydon's book, *The spectroscopy of Flames* (1957), he mentions that the CaOH bands are 'a persistent impurity in flame spectra, especially in explosion flames' (p354). It is curious that these compounds of calcium are the ones seen, and not the pure element themselves. It is something we don't have an explanation for, but they are the ones that consistently appear in combustion data.

This is also an area which is not always well known – it is relatively common to assign such lines to other elements that one would *assume* to be present in the molecule, rather than considering impurities. Bringing more attention to the common emissive compounds seen in combustion spectra would be useful in lessening the amount of wrongly assigned lines of which we have numerous examples (this can be seen in a recently published paper¹ on flame spectra, if you're interested).

1. Goyal, T., Trivedi, D. and Samimi Abianeh, O. Autoignition and flame spectroscopy of propane mixture in a rapid compression machine. *Fuel* **233**, 56-67 (2018).

10. Small periodic structures in figure 1

This is an artefact present in my spectrometer, which we rightly should have mentioned. We thank you for your observation. It is more often seen at lower intensities, of where the calibration failed to completely remove the dark current present in equipment. However, we have taken full account of this in my analysis. In reviewing the manuscript, we have changed the figure to reflect the data post-analysis.

I hope this clarifies the aim of the paper, and answers the questions you had. Please let me know if you require anything else, or have a suggestion to make. It is incredibly useful to have input from experts about this topic – especially the theory – as it is something that is underappreciated in many communities.

Dear Reviewer 3,

Thank you for your time and consideration in reviewing this manuscript.

Before we go point-by-point replying to your queries, we would first like to draw your attention to the fact that, after further analysis in the interim (and taking account of an offset present we had not previously accounted for), we have re-calibrated the temperature involved to result in slight difference to those previously reported. For that reason, the Hopkinson bar temperatures are now reported as 2900 K, rather than 2500 K, and the gradient has changed to 950 as a result. This has given us the opportunity to re-do all our calculations and we are satisfied everything is now correct.

1. Extend to higher pressures / look at DDT.

This is definitely an area of research that we want to study more closely. Especially in a system such as DDT. For instance, the presence of a streak spectrometer on DDT may allow a clearer view of the peak shift as the reaction pressures increase, as well as providing temperature measurements.

So it is certainly a plan for future research, but outside the scope of this current study. The purpose of which is to draw the readers' attention to the presence of sodium, and how it can be exploited, then focus onto something specific for HMX – could equally have used RDX or PETN.

2. SHPB used on HMX before.

We were aware that the mechanical properties of HMX had been studied in a SHPB. As you pointed out, in those studies initiation is seen as something to be avoided, so it is not as widely reported in the academic community. Here the deliberate purpose was the study of initiation.

Our thanks for the link to the APS paper – it is interesting to note they achieved initiation but not a complete reaction. Even with loose powder, you require larger velocities than normally achieved in the Hopkinson bar to initiate. It is potentially another area of further study, to investigate IM and also pre-damaged materials and their initiating pressures in a SHPB apparatus.

3. Figure 3 hard to see point

We agree, especially given how the error bars are small for that point. We referenced that it was at the origin in the caption below, and have now added its position into the legend to further clarify.

Thank you for your comments – this is the first look at a new diagnostic, but I am hoping to apply it to more energetic systems during my PhD studies (I have just completed year 1). As an expert in your field, your queries/suggestions are always appreciated.

Reviewers' comments:

Reviewer #1 (Remarks to the Author):

The authors have sufficiently responded to the questions and comments of the reviewers and made appropriate changes to improve the manuscript. I recommend it for publication at this point.

Reviewer #2 (Remarks to the Author):

In the revised manuscript, the authors responded reasonably to many issues raised in my first review. The most important point that needs a further clarification is related to the Stark shift.

It is stated that "... an electric field of order $10 \text{ kV}\cdot\text{cm}^{-1}$ can be estimated to be present in the reaction [...] result in a shift of order $2\cdot 10^{-5} \text{ nm}$, which is negligible compared to the collisional effects".

However, the DC Stark shift is not what I had in mind. Instead, the collisions with charged perturbers (electrons and ions) cause not only the Stark line broadening, but the shift as well, and these are often comparable, especially in the low-temperature (by plasma-physics standards) region. For the Na D line, the Stark shift would be around 1 nm for $n_e = 10^{19} \text{ 1/cm}^3$ (see, e.g., <https://griem.obspm.fr>, <http://stark-b.obspm.fr>). It is not evident for me that the conditions of the experiments described in the study preclude such a modest level of ionization. Indeed, at $T = 3000 \text{ K}$, a $\sim \text{GPa}$ pressure means $\sim 10^{22} \text{ 1/cm}^3$ neutrals, give or take. So the ionization degree of 0.1% would already be very important in terms of the line shift interpretation. Please evaluate the density of charged particles under conditions of your experiments based on the Saha-Boltzmann equation.

Reviewer #3 (Remarks to the Author):

The authors appear to have nominally addressed my previous comments.

Reviewer 2 is correct for a pure sodium environment – that the ionization level is high. However, in this particular system, sodium is a minor impurity in the material, and the overwhelming majority of perturbers are the reaction products (N₂, H₂O and CO₂). These have higher ionization energies compared to sodium, and so when applying the Saha-Boltzmann equation, lead to significantly lower charge densities present.

A back-of-the-envelope calculation, using the average ionization energies of these species (14, 15.6, 12.6 eV), gives an electron density of *circa* $5 \cdot 10^{15} \text{ cm}^{-3}$ (Saha-Boltzmann equation) at a temperature of 4000 K. This density, using the data from <http://stark-b.obspm.fr/>, gives a shift of order 10^{-4} nm, which is negligible compared to the observed results.

Independently, given the temperatures and pressures present in this research, one would expect an increase of temperature to lead to a larger Stark red-shift (direct relationship). Whereas the experimental data fits the functional form of the collisional theory, with a smaller shift at higher temperatures (inverse relationship).

We have included a sentence with this order-of-magnitude explanation alongside the DC Stark shift discussion.

REVIEWERS' COMMENTS:

Reviewer #2 (Remarks to the Author):

"Independently, given the temperatures and pressures present in this research, one would expect an increase of temperature to lead to a larger Stark red-shift (direct relationship). Whereas the experimental data fits the functional form of the collisional theory, with a smaller shift at higher temperatures (inverse relationship)."

No, it is the same direct relationship; have a look at your Eq. (1), giving $\sim T^{3/10}$. Are you confusing it with the isobaric dependence, Eq. (2)? The Stark-B data are for a constant density, not pressure.